# Temporal Analysis of *Candidatus* Liberibacter asiaticus in Citrandarin Genotypes Indicates Unstable Infection



**Thais Magni Cavichioli** [1,†] **, Maiara Curtolo** [1] **, Mariangela Cristofani-Yaly** [1] **, Josiane Rodrigues** [2] **and Helvécio Della Coletta-Filho** [1,*]

[1] Centro APTA Citros Sylvio Moreira, Instituto Agronômico de Campinas, Cordeirópolis 13492-442, SP, Brazil
[2] Department of Tecnologia Agroindustrial e Socioeconomia Rural, Univeridade Federal de São Carlos, Araras 13600-970, SP, Brazil
* Correspondence: hdcoletta@ccsm.br; Tel.: +55-(19)-3546-1399
† Unesp, FCAV, Jaboticabal 14884-900, SP, Brazil.

**Abstract:** Background: *Huanglongbing* (HLB) is currently one of the most devasting diseases in citrus plants worldwide. Resistance against its causal agent, *Candidatus* Liberibacter asiaticus (*C*Las), in commercial *Citrus* species remains a challenge, even though they show differences in *C*Las multiplication. Methods: A total of 14 citrandarins and their parents (Sunki mandarin and *Poncirus trifoliata* cv. Rubidoux) were top-grafted onto the canopy of potted 'Valencia' sweet orange plants with high *C*Las titers. The grafted genotypes were tested for *C*Las infection and physiological effects of the disease (starch accumulation and callose deposition) for 12 months. Results: All tested genotypes were infected by *C*Las during the time frame of the experiment. However, a decrease in the infection rate ranging from 50% to 80% for the hybrids H109, H126, H157, and H222 was observed 360 days from the top-grafting. *C*Las was undetected by real-time PCR in H106 at the end of the experiment, which had low levels of starch and callose deposition. Conclusions: *C*Las infected all of the tested citrandarins, but a decrease in the infection rate over time was detected for some specific genotypes, which led to less starch accumulation and callose deposition.

**Keywords:** Huanglongbing; citrus rootstock; disease tolerance; bacteria





## 1. Introduction

Huanglongbing (HLB) has been considered one of the most important diseases occurring in citrus plants worldwide, causing severe economic losses in commercial production [1]. The Gram-negative bacterium *Candidatus* Liberibacter asiaticus (*C*Las), vectored by the insect *Diaphorina citri* [2], is the main pathogen associated with HLB. The bacterium colonizes the phloem vessels of the host plant, causing symptoms such as chlorotic mottling of leaves, small and deformed fruits, fruit drop, aborted seeds, and bitter juice [2]. The decline of plants due to *C*Las infection is attributed to a series of physiological, cellular, and molecular disorders, such as phloem collapse, disintegration of the root system, and drastic disruption in the translocation of carbohydrates [3]. The latter includes starch accumulation in leaves and callose deposition in the phloem [4,5], which negatively affect the carbohydrate distribution for sink tissues (roots or flushing leaves) [6].

The management of HLB comprises complementary actions, such as the exclusion of the pathogen by the use of healthy propagating material, the eradication of the *C*Las-inoculum source by uprooting the infected plants, and the application of insecticides to suppress the vector [7]. Even though the mentioned strategies are technically well-supported, adoption by growers has decreased over time, likely for economic reasons. The use of fewer HLB-susceptible varieties, such as those used as rootstocks, has been discussed as a complement aiming to increase the management efficiency as well as to reduce the disease impact [8].

No CLas-resistant commercial citrus genotypes have been reported in the literature so far. However, the multiplication of CLas differs among *Citrus* species belonging to the tribe Citreae [9,10]. *Poncirus trifoliata* (L. Raf.) and some of its hybrids exhibited tolerance to the bacteria [11–14]. In addition, some hybrids of *P. trifoliata* used as citrus rootstocks can increase scion fruit quality by improving the total soluble solids content and contributing to reduced plant size [15,16].

Previous studies from our group have shown differences in the multiplication of CLas in the progeny of citrandarins, hybrids of *P. trifoliata* cultivar Rubidoux (Pt cv Rub), and Sunki mandarin [*Citrus sunki* (Hayata) hort. ex Tanaka], as well as in their parents [14,17]. Field experiments in Cordeirópolis—São Paulo, Brazil (an area with high CLas inoculum pressure) using 275 citrandarin genotypes grafted as scions on the Rangpur lime (*C. limonia* Osbeck) rootstock revealed that some genotypes were not infected by the pathogen through natural transmission [17]. However, these authors could not exclude the failure of inoculation by the vector. Next, some hybrids selected based on the absence of CLas detection were challenged with CLas via infected buds, and any bacteria were detected 12 months after grafting [13,14]. In addition to differences in the infection rate and CLas titer, the authors also revealed variations in the accumulation of starch and callose deposition among the genotypes, indicating differential responses to CLas infection. Despite the evidence of resistance against CLas, the transmission of bacteria by grafting vegetative material from HLB-infected plants still varies due to the irregular distribution of bacteria in the infected plants [18,19].

This work aimed to investigate the colonization of CLas in 14 different citrandarins regarding their HLB resistance. For that, those different genotypes were grafted on sweet orange Valencia plants with HLB symptoms and high CLas titers. These genotypes were previously identified as resistant to CLas because no bacteria were detected in the plants [14, 17], or were selected based on their horticultural advantages, such as fruit yields and drought tolerance (Christofany-Yale, personal communication). CLas frequencies and titers in the grafted plants were determined by qRT-PCR, and physiological responses to the infection (i.e., starch accumulation and callose deposition) were examined for 12 months. The tested genotypes showed no resistance to CLas. However, after 12 months from inoculation, some genotypes showed a reduction in the percentage of CLas-infected replicates, including no detection of bacteria in all reps of one of the tested genotypes, which characterizes a transient infection in these genotypes.

## 2. Materials and Methods

### 2.1. Plant Material and CLas Infection

To evaluate the resistance of citrandarin genotypes against CLas, 14 hybrids (Table 1) were selected based on previous information such as: (i) no natural infection by CLas after 12 years in field experiments (H106, H222, and H199-[17]); (ii) no infection by graft inoculation (H68 and H106-[14]); and (iii) lack of information for hybrids with no HLB symptoms and no CLas detection, even when these genotypes were cropped for 12 years in field conditions under high HLB incidence. Five-centimeter budsticks from the 14 hybrids and the parents Sunki mandarim and *Poncirus trifoliata* cv Rubidoux (Pt cv Rub), whose mother plants had been maintained in a greenhouse, were top-grafted on three-year-old HLB-symptomatic sweet orange Valencia scion grafted on Rangpur lime (*Citrus X limonia* hybrid) grown in 30-litre-pots containing commercial substrate PlantMax® (pine bark, charcoal, vermiculite, lime, and nutrients—NPK and micronutrients) from Terra do Paraiso company (Holambra, Sao Paulo, Basil). Branches from the HLB-affected Valencia plants that received budsticks from the selected genotypes (citrandarins and parentals) were confirmed for the presence of CLas by real-time quantitative PCR (qPCR). All grafted budsticks were protected with a small plastic bag for 30 days after the top-grafting to maintain high humidity and improve the grafting success (Figure S1). Each Valencia HLB-diseased plant received from four to six different genotypes, and a total of six replicates of each genotype were grafted. Unfortunately, failure in the grafting process led to a

variable number of replicates ($n$ = 3 to 6) for each genotype. The Valencia HLB-diseased plants containing the top-grafted plants were randomly distributed in the greenhouse under natural lighting conditions and the temperature did not exceed 30 °C during the entire experimental period. All plants were automatically irrigated twice daily (0.8 L of water/plant/day) and weekly fertilized with a solution containing N: $P_2O_5$: $K_2O$ and micronutrient complex. Phytosanitary treatments, mainly for spider mite control, were done when necessary.

**Table 1.** $C_T$ values of different Valencia HLB-diseased plants infected by *Candidatus* Liberibacter asiaticus (*C*Las) that received the citrandarin genotypes (hybrids and parents) by top-grafting. Plants were analyzed for 12 months.

| Valencia HLB-Diseased Plants (VP) | Grafted Genotypes [1] | Days After Top-Grafting [2] | | | |
|---|---|---|---|---|---|
| | | 90 | 180 | 270 | 360 |
| VP02 | 126, 158, 181, 303, 106 | 24.98 Bb | 22.72 Bc | 24.65 Ab | 26.95 Aa |
| VP05 | 109, 199, 222, 254, 303 | 27.70 Aa | 22.10 Bc | 23.87 Bb | 24.53 Ab |
| VP11 | 158, 282, Pt, 75, 303 | 21.00 Cb | 21.41 Cb | 20.82 Cb | 23.06 Ba |
| VP14 | 75, 109, 126, 222, 282, Pt | 24.16 Ba | 24.84 Aa | 25.37 Aa | 25.40 Aa |
| VP16 | 158, 283, Pt, 303, 75 | 22.08 Cb | 22.63 Bb | 25.54 Aa | 26.01 Aa |
| VP19 | 181, 222, 303,109, 181 | 22.93 Ca | 23.89 Aa | 22.98 Ba | 21.08 Bb |
| VP20 | 199, 254, 283, 106, Sk | 22.51 Ca | 19.41 Cb | 21.54 Ca | 21.81 Ba |
| VP21 | 106, 199, 282, 68, Sk | 21.31 Ca | 20.82 Ca | 19.33 Cb | 21.74 Ba |
| VP22 | 68, 106, 222, 109, 158 | 21.57 Ca | 22.35 Ba | 21.81 Ca | 21.23Ba |
| VP23 | 68, 109, 303, Sk, Pt | 23.11 Ca | 21.24 Cb | 20.66 Cb | 23.00 Ba |
| VP24 | 126, 157, 254, 283, Sk | 21.70 Ca | 20.42 Ca | 20.66 Ca | 21.59 Ba |
| VP25 | 68, 75, 157, 254, 283, Pt | 19.44 Da | 20.29 Ca | 20.71Ca | 19.89 Ba |
| VP26 | 106, 157, 222, 254, 283, Sk | 18.98 Dc | 20.81 Cb | 23.54 Ba | 21.48 Bb |
| VP27 | 68, 106, 282, 283, 109 | 20.58 Da | 21.21 Ca | 22.75 Ba | 21.68 Ba |
| Median/*C*Las titer [3] | | 21.89 4.91 × $10^5$ | 21.32 7.41 × $10^5$ | 22.28 3.70 × $10^5$ | 21.77 5.35 × $10^5$ |

[1] Numbers refer to the identification of citrandarins. Pt and Sk are the parentals *Poncirus trifoliata* cv and Rubidoux and Sunki mandarin, respectively. Numbers represent average of $C_T$ values of branches on Valencia plants where the tested genotypes were grafted. Significant differences (Scott-Knott test for comparison of the means, $p < 0.05\%$) are indicated by letters. Small letters (within lines) compare values from different days after grafting. Capital letters (within columns) compare different Valencia HLB-diseased plants. [3] Based on the 16S rDNA *C*Las copy number per g of fresh tissue.

## 2.2. Sampling

Completely expanded leaves of the tested genotypes showing dark green color were harvested at 90, 180, 270, and 360 days after top-grafting. Petioles and leaf blades were used for *C*Las and starch quantification, respectively. The same parameters were determined in Valencia HLB-diseased plants.

## 2.3. DNA Extraction and RT-qPCR Analysis

Aliquots (200 mg of petiole slices from hybrids, parentals, and Valencia HLB-diseased plants) were used for DNA extraction. In summary, the petioles were homogenized in plastic tubes containing two 5 mm diameter tungsten spheres using the TissueLyser II (Qiagen, São Paulo, Brazil), followed by DNA extraction with CTAB [20]. The total DNA was eluted in 200 µL of 1/10 TE + RNAse.

The diagnosis and quantification of *C*Las were performed using qPCR based on the protocol of Li et al. [21]. The amplifications were performed in duplex using an ABI Prism 7500 Sequence Detection System thermocycler (Applied Biosystem, Walthan, MA, USA). The reaction contained 6.25 mL of 1X TaqMan Universal Master Mix (Applied Biosystem), 216 nM of each primer, and 135 nM of the *C*Las (HLBp) probe, for a total volume of 14 µL. Primers and probes targeting the glyceraldehyde 3-phosphate dehydrogenase (*GAPDH*) gene of citrus were used as internal controls, with 270 nM of each primer and 135 nM of the probe [17]. Aliquots (3 µL) of total DNA standardized at 100 ng/µL were added per reaction. The amplification cycle followed the default settings of the equipment: 95 °C for 5 min, followed by 40 cycles of 95 °C for 30 s, and 58 °C for 45 s.

### 2.4. Determination of the Cycle Threshold ($C_T$) for the Detection and Quantification of CLas

To determine the threshold $C_T$ value used to infer whether the plants were *C*Las-positive (infected) or *C*Las-negative (free from the pathogen), a standard curve for estimating the bacterial titer based on the 16S rDNA content of *C*Las [21] was generated according to Wang et al. [22] (Figure S2). The target fragment was cloned into the recombinant plasmid using the pGEM—T-Easy Vector System (Promega Corporation), sequenced by the Sanger method, and blasted against the NCBI database (https://blast.ncbi.nlm.nih.gov/ accessed on 30 June 2020), using the BLASTn tool to certify the origin of the amplicon. The bacterial population was measured as the copy number of the 16S rDNA target gene $g^{-1}$ of tissue using serial dilutions of the plasmid DNA (from $3.0 \times 10^9$ to 0.3 molecules 16S rDNA copies/µL). The standard curve obtained shows a linear regression (y = $-3.1828x + 40.004$) with a very high correlation between the variables ($R^2 = 0.9994$) and with efficiency (E) of 1.06 (Figure S2).

### 2.5. Starch Quantification

The enzymatic quantification of starch in all tested plants was carried out using 10 mg of dry plant material according to Amaral et al. [23]. Briefly, soluble sugars were extracted with 80% ethanol three times and the remaining pellet was treated with α-amylase and amyloglucosidase. Aliquots of 20 µL were incubated with glucose released by the enzymatic digestion of starch and quantified with the commercially available Glucose PAP Liquiform kit (Centerlab, Lagoa Santa, MG, Brazil) using a microplate reader (Model 3550—BIO-RAD) at 490 nm (Figure S3).

### 2.6. Callose Visualization and Quantification

The quantification of callose in leaf petioles was carried out at 360 days from inoculation using 10 cross sections 0.02 mm thick obtained by an automatic slide microtome (Leica SM2010R). Samples were stained with blue aniline at a concentration of 0.01% in 0.1 M phosphate buffer (pH 9.0) [24] and imaged using an Olympus BX61 fluorescence microscope at a magnification of $10\times$, with a 355–375 nm excitation filter, a 400 nm dichromatic mirror, and a 435–490 nm emission filter. Callose was quantified from the fluorescence intensity emitted by the stain (pixel/mm$^2$) using ImageJ software (Version 1.45 s) (https://imagej.nih.gov/ij/ downloaded on 24 March, 2021) according to Granatto et al. [25].

### 2.7. Statistical Analysis

The Scott–Knott test was carried out at 5% significance using R software. For the percentage dataset, the arc-sin transformation was applied before the analysis.

## 3. Results

### 3.1. CLas Detection and Quantification in Valencia HLB-Diseased Plants

Based on interpolation of the $C_T$ value and the number of copies of *C*Las 16S rDNA added to the reaction, the minimum amount of the *C*Las-16S rDNA insert that was reproducibly amplified was equivalent to approximately 90 copies of the 16S rDNA molecule. The calculations took into account the presence of three copies of 16S rDNA in the *C*Las genome [26], equivalent to $C_T = 31.73 \pm 0.75$ (Table S1). Amplifications using total DNA from negative controls (micrografted citrus plants kept in a greenhouse) showed $C_T$ values ranging from "undetermined" to 33.29 (average of $34.53 \pm 1.24$). As the $C_T$ values are inversely proportional to the concentration of bacteria in the sample, under our experimental conditions, samples were considered positive for *C*Las when $C_T < 32$ and negative when $C_T > 32$. Amplifications with a $C_T$ value close to 32 ($\pm 1$) were repeated and the results were considered only if reproducible.

All the Valencia HLB-diseased plants, including the specific branches that received the grafted budsticks (citrandarins and parents), were already infected with *C*Las at the time of the top-grafting, and the plants showed regular HLB symptoms such as leaf mottling (data not shown). In general, *C*Las titer varied significantly through the experimental period

within the same Valencia plant, and the maximum absolute range was 3.93 $C_T$ units for the Valencia HLB-diseased plant SP16, followed by the SP05 and SP26 with 3.17 and 2.5 $C_T$ units, respectively (Table 1). Variation in *C*Las titer among the Valencia plants was also detected (Table 1) but with a similar median for the $C_T$ values among the sampling points, indicating a narrow range of bacteria titer (16S rDNA *C*Las copy number from $3.70 \times 10^5$ to $7.41 \times 10^5$ per g of tissue). Significant deviation from the median was observed in a few *C*Las Valencia plants such as for the sampling points SP05 at 90 days after top-grafting, SP02 and SP16 at 360 days after top-grafting (Table 1).

### 3.2. Temporal Infection of CLas in Top-Grafted Genotypes

$C_T$ values at the first sampling time (90 days after top-grafting) showed that in the majority of the 14 hybrids, 50% of their replicates were positive for *C*Las infection. However, in the subsequent sampling times, a decrease in the number of infected replicates was observed for H109, H126, H157, H222, and the *Pt* cv Rub parent. Additionally, no *C*Las-infected replicate was observed for H106 360 days after top-grafting (Table 2). The four groups of hybrids were significantly different from each other based on the frequency of *C*Las-infected replicates. H106 was unique because it had no *C*Las-infected replicates at the end of the experiment. The second group had a frequency of positive replicates ranging from 25% (H222) to 60% (H68), and was composed of H68, H75, H109, H126, H157, H181, H222, H303, and the *P. trifoliata* parent. H254 and H283 formed the third group, with 80% of replicates infected with *C*Las, while H158, H199, H282, and the Sunki mandarin parent formed the fourth group, with 100% of *C*Las-positive replicates (Table 2).

**Table 2.** Number of *C*Las-infected shoots from citrandarins and parents top-grafted onto HLB-diseased Valencia plants.

| Genotypes | Days After Top-Grafting [1] | | | |
|---|---|---|---|---|
| | 90 | 180 | 270 | 360 |
| Hybrids | | | | |
| H68 | 4/5 [2] b | 2/5 c | 4/5 b | 3/5 c |
| H75 | 1/4 c | 2/4 c | 2/4 b | 2/4 c |
| H106 | 3/4 b | 3/4 b | 3/4 b | 0/4 d |
| H109 | 5/6 b | 2/6 c | 1/6 c | 2/6 c |
| H126 | 2/3 b | 2/3 b | 2/3 b | 1/3 c |
| H157 | 3/3 a | 1/3 c | 1/3 c | 1/3 c |
| H158 | 4/4 a | 4/4 a | 4/4 a | 4/4 a |
| H181 | 2/3 b | 2/3 b | 2/3 b | 2/3 c |
| H199 | 3/3 a | 3/3 a | 3/3 a | 3/3 a |
| H222 | 5/5 a | 3/5 b | 3/5 b | 1/5 c |
| H254 | 5/5 a | 4/5 b | 3/5 b | 4/5 b |
| H282 | 4/4 a | 4/4 a | 4/4 a | 4/4 a |
| H283 | 5/6 b | 4/6 b | 4/6 b | 4/5 b |
| H303 | 4/6 b | 4/6 b | 3/6 b | 3/4 b |
| Parents | | | | |
| Sunki mandarin | 5/5 a | 5/5 a | 5/5 a | 5/5 a |
| *P. trifoliata* cv Rubidoux | 3/6 c | 2/6 c | 2/5 b | 2/5 c |

[1] Plants were harvested within one year after the top-grafting and *C*Las propagation was analyzed by RT-qPCR. The mean $C_T$ values in those plants can be seen in Table 1. [2] Number of positive hybrids/ total number of grafts. Dead grafts led to reductions in the number of analyzed samples in relation to previous sampling times. Significant differences (Scott–Knott testing for comparison of the means) among the genotypes are shown by letters.

Despite the decrease in the number of infected replicates over time, the concentration of *C*Las in all positive replicates of the hybrids and their parents, based on $C_T$ values, was similar during the whole experimental period, with a few exceptions at 360 days after the top-grafting (Table 3). We could distinguish two groups, with lower and higher $C_T$ values. H283, H282, H222, H199, H157, H68, and the *C*Las-susceptible parent Sunki mandarin belong to the first group, with a *C*Las titer reaching $8.84 \times 10^5$ copy number/g tissue.

Another citrandarin group (H75, H106, H126, H158, H181, H254, H303) and the tolerant parent Pt cv Rub had a *CLas* titer reaching a maximum of $9.94 \times 10^3$ copy number/g tissue (Table 3). A sudden decrease of *CLas*-infection was observed for H106 at 360 days after top-grafting, with no *CLas* detection in all replicates (Tables 2 and 3). As previously mentioned, during whole the experimental period, all tested plants were top-grafted over HLB-diseased Valencia plants constantly having a high volume of *CLas*, from 3.7 to 7.41 $\times$ $10^5$ copy number of bacteria/g tissue (Table 1).

**Table 3.** $C_T$ values and concentrations of *Candidatus* Liberibacter asiaticus (*CLas*) in citrandarins. Plants were analyzed for 12 months and a decrease in the number of replicates positive for *CLas* infection was observed over time.

| Genotypes | Days After Top-Grafting (DATg) [1] | | | | CLas Titer at 360 DATg [2] |
|---|---|---|---|---|---|
| | 90 | 180 | 270 | 360 | |
| | | | Hybrids | | |
| H68 | 28.92 a | 25.55 a | 23.66 a | 23.37 b | $1.68 \times 10^5$ |
| H75 | 19.72 c | 23.28 a | 20.76 a | 28.65 a | $3.69 \times 10^3$ |
| H106 | 29.40 a | 25.02 a | 26.05 a | 40.00 [3] | no-detected cells |
| H109 | 25.96 a | 26.62 a | 25.42 a | 27.28 a | $9.95 \times 10^3$ |
| H126 | 25.39 a | 25.15 a | 30.13 a | 25.74 a | $3.03 \times 10^4$ |
| H157 | 27.71 a | 23.64 a | 24.18 a | 23.10 b | $2.05 \times 10^5$ |
| H158 | 23.45 b | 24.40 a | 25.81 a | 27.59 a | $7.94 \times 10^3$ |
| H181 | 26.63 a | 26.68 a | 28.25 a | 29.05 a | $2.76 \times 10^3$ |
| H199 | 24.77 b | 23.70 a | 21.17 a | 22.95 b | $2.28 \times 10^5$ |
| H222 | 27.15 a | 25.98 a | 24.31 a | 24.93 b | $5.44 \times 10^4$ |
| H254 | 28.00 a | 27.28 a | 28.09 a | 27.28 a | $9.94 \times 10^3$ |
| H282 | 25.58 a | 24.46 a | 22.46 a | 23.31 b | $1.76 \times 10^5$ |
| H283 | 23.02 b | 23.93 a | 21.47 a | 21.14 b | $8.84 \times 10^5$ |
| H303 | 27.29 a | 26.95 a | 22.66 a | 27.68 a | $7.44 \times 10^3$ |
| Parents | | | | | |
| Sunki mandarin | 27.50 a | 25.50 a | 24.02 a | 23.93 b | $1.12 \times 10^5$ |
| *P. trifoliata cv Rubidoux* | 28.93 a | 25.69 a | 26.31 a | 29.99 a | $1.40 \times 10^3$ |
| CV (%) | 11.67 | 11.39 | 11.97 | 10.03 | |

[1] Significant differences (Scott–Knott testing for comparison of the means, *p < 0.05%*) in a genotype among sampling times are shown by letters. Numbers represent the average $C_T$ values of *CLas* present in sprouts of citrandarin genotypes and the parents. [2] Copy number of *CLas* 16S rDNA target gene per gram of fresh tissue. [3] Not included in the statistical analysis.

### 3.3. Physiological Response of Citrandarins to CLas Infection Based on Starch Accumulation and Callose Deposition

HLB symptom expression in P. trifoliata and seedlings suchs as citrandarin genotypes is often inconsistent probably due to seedlings variation. The same is valide for determination of sprout lenght of the graphed citrandarins during the experiment based on know variation of vigor within the citrandains poopulations. Therefore starch accumulation and callose deposition was quantified as a measure of the response to CLas infection. Starch accumulation in the tested plants was significantly lower than in the corresponding Valencia HLB-diseased plants. The exception was the susceptible parent Sunki mandarin, as this difference was not significant (76.41 and 87.27 mg glucose/g of dry tissue in Sunki and Valencia, respectively). However, the average starch content in the 14 citrandarins was significantly lower compared to the parent Sunki. Parent Pt cv Rub had lower starch (12.41 mg glucose/g of tissue), similar to genotypes H109, H126, H181, H199, H222, H282, and H303 (12.41 to 20.88 mg glucose/g dry tissue) (Figure 1A). Genotypes with a starch content ranging from 26.23 to 49.11 mg glucose/g of dry tissue (H68, H106, H158, H157, H254, and H282) had intermediate levels of starch compared to parents Sunki and Pt cv Rub. Hybrids with a high frequency of infection, such as H181, or with no detectable bacterium (H106) 12 months after the top-grafting (Table 2) accumulated similar amounts of starch (Figure 1A). Based on the frequency of infection and starch accumulation, parents and some hybrids were selected for the determination of callose deposition (Figure 1B). The parent Pt cv Rub, considered resistant, showed significantly less callose deposition in the petiole,

followed by H106. Hybrids with a high (H199 and H282) or lower (H157) frequency of CLas-positive replicates showed similar values for callose deposits that were significantly lower than those observed for Sunki mandarin, the susceptible genotype (Figure 1B).

A

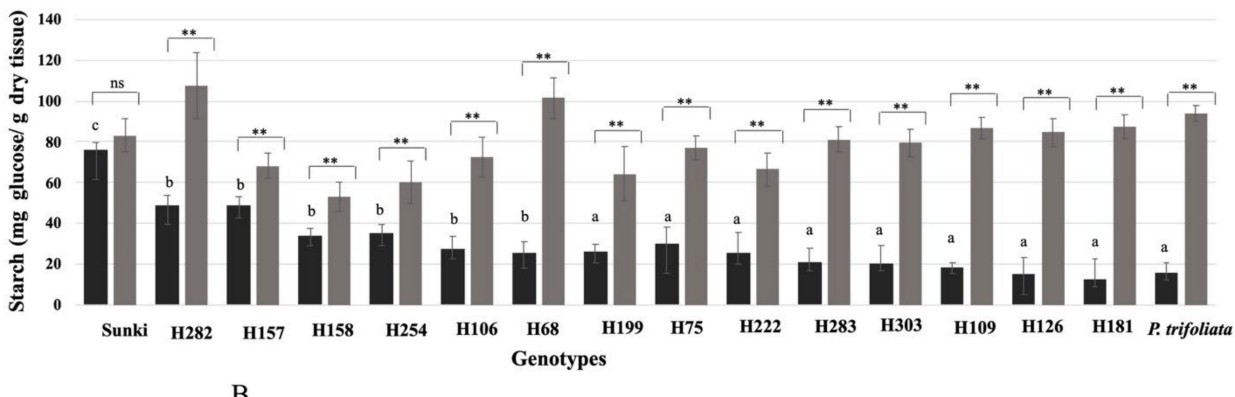

B

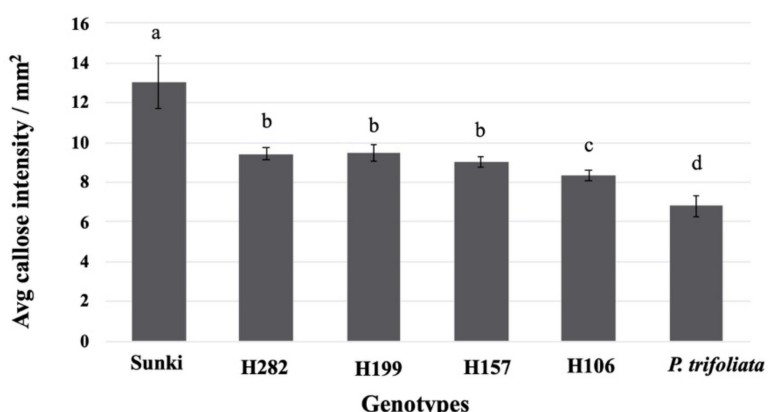

**Figure 1.** Physiological responses of citrandarins and parent species *C. sunki* and *Poncirus trifoliata* to CLas infection. (**A**) Average starch content obtained from four sampling points (90, 180, 270, and 360 days from top-grafting). Vertical bars represent standard deviations of replicates for each genotype across sampling dates. Significant differences (Scott-Knott testing for comparison of the means) among citrandarins are shown by letters. Significant differences (*t*-testing for comparison of the means) between hybrids or parents (black bars) and the Valencia HLB-diseased plants (gray bars) are indicated by asterisks (** $p < 0.001$); NS—no significant. (**B**) Callose quantification in leaf petioles of selected genotypes. Vertical bars represent standard deviations of 10 cross-sections for each genotype sampled 12 months after top-grafting. Significant differences (Scott–Knott testing for comparison of the means, $p < 0.05$) among citrandarins are shown by letters.

## 4. Discussion

There is still no effective method to control CLas in the host plants and, consequently, the HLB. The current strategy of HLB management is based on planting healthy plants, controlling the vector (psyllids) using insecticides or biological agents, and removing Clas inoculum sources (infected trees) [1]. Strategies aiming to reduce the infection or increase the plant response against the disease by supplying antimicrobial molecules based on plant metabolites, nutrients, and biostimulants applied to soil [27–30] showed variable beneficial results. These studies have been carried out on scion-rootstock combinations with known susceptibility to CLas infection and showing HLB symptoms. Although CLas can infect most *Citrus* and relative species, some are more tolerant to HLB than others. For

instance, *C. sinensis*, *C. reticulata*, *C. paradisi*, *C. maxima*, and *C. aurantifolia* are CLas-sensitive species [31–33], whereas *P. trifoliata* and some hybrids are considered more tolerant, as also are other citrus relatives such as *Microcitrus* spp. and *Eremocitrus glauca*. Variations within these more tolerant species have been reported [10,34]. *P. trifoliata* and some of its hybrids have received special attention concerning the lower and/or variable CLas titer detected [11,17,31] and milder HLB symptoms and physiological responses to CLas infection [11,13].

In this work, we detected no widespread resistance against CLas in the studied geno-types, which is in agreement with previous reports [11,13,14,17,35]. However, many different selections of these species exist, and they have shown variable responses to CLas [34], as also observed in this work (Tables 2 and 3). Interestingly, the frequency of CLas-positive replicates dropped over time for some hybrids (H109, H126, H157, and H222). For example, H222 and H106 had 100% of infection at 90 days after top-grafting, which fell to 20% and zero (i.e., no CLas detection) at 360 days after the top-grafting, respectively. CLas was also not detectable in the scion of the hybrid H106, even by graft inoculation [14] or natural infection by the vector [17]. CLas infection was detected in H106 under our experimental setup (high CLas titer in the Valencia HLB-disease plants), which, from our point of view, minimizes possible failures of inoculation by the constant flux of CLas from the Valencia HLB-disease (source of CLas) plant to the pathogen-free grafted genotypes [36] (Figure S1, Table 1). Following this hypothesis, H68 previously reported as noninfected by graft inoculation [14], had 65% of CLas-positive replicates (Table 2). The grafting transmission of pathogens can fail due to the irregular distribution of pathogens in the infected tissue [37]. Furthermore, successful grafting involves complex processes including the initial anatomical deformities of vascular tissue followed by the time-dependent connection and differentiation into xylem and phloem at the graft junction [38], which could impair vascular pathogen transmission. These anatomical deformities could explain CLas-positive budsticks and noninfected grafted plants 18 months after inoculation [13]. Our top working strategy was used by Fadel et al. [39] to demonstrate the tolerance of sweet orange Navelina Iso 315 against *X. fastidiosa* bacterium.

Starch accumulations in leaves and callose deposition in phloem have been positively associated with host susceptibility to CLas [13,25,40–43]. Starch content in the Valencia HLB disease branches (CLas-positive) and in the susceptible Sunki mandarin parent were similar and statistically different from citrandarins and Pt cv Rub (Figure 1A). All citrandarins and Pt cv Rub whether infected or not by CLas accumulated significantly less starch in comparison with Sunki or Valencia plants, but with variation among the citrandarin geno-types and the parental *P. trifoliata* (Figure 1A). These results reinforce previous information about the variation in starch levels in *P. trifoliata* and hybrids under CLas infection [13,14]. Nonetheless, parameters such as the frequency of CLas-infected replicates and bacteria titer could not be directly associated with the starch content, as exemplified by the hybrids H157, H181, H199, H283, and H303 (Tables 2 and 3, Figure 1A). Specifically, variation in the starch content in H106 throughout the experiment was observed, from 35.78 ($\pm$1.77) mg glucose/g of tissue at 90 days after top-grafting to 20.15 ($\pm$0.64) at the end of the experiment, when CLas could not be detected. We believe that the previous selection of these hybrids based on higher tolerance to HLB could explain this low starch content, even in the presence of CLas. Callose deposition was also lower in H106, reinforcing its better response against the effects of CLas infection, as reported by Curtolo et al. [14] (Figure 1B). Our results corroborate previous findings on fewer pronounced physiological events associated with CLas infection, such as massive starch accumulation in leaves, excessive callose deposition in sieve pores of phloem, and imbalance of carbohydrates within the plant [42] in *P. trifoliata* [13,14].

Explaining the probable mechanism of tolerance to HLB of some *P. trifoliata*-related genotypes, and in a few cases, to CLas (e.g., H106), is still in the early stages. Genes related to bacterial cell lysis, such as *endochitinase B* [44], were upregulated in the pool of CLas-resistant citrandarins [14]; as well, candidate genes for CLas tolerance such as the transcriptor factor *WRKY70* and two *nucleotidebinding site (NBS)-leucine-rich repeat (LRR)*

clusters were identified in *Poncirus* genotypes [45,46]. In addition, stable antimicrobial peptides (SAMPs) found in *Microcitrus australiasica*, which were shown to effectively reduce *C*Las titer, HLB symptoms, and induce innate immunity to prevent and inhibit *C*Las infections, were found equally expressed in *P. trifoliata* cultivar Texas, but not in the other trifoliata cultivars tested [47].

The use of more tolerant rootstocks could be one additional strategy against the deleterious effects of HLB in citrus plants, including the ones affecting the root system [6]. In this work, we showed that *C*Las was detected in all tested hybrids, suggesting no resistance to this pathogen. However, variation on response to *C*Las infection based on starch accumulation, calloses deposition, and bacteria concentration has emerged, including a decrease in infection ratio in some hybrids (H109, H126, H157, and H222), mainly in H106. This was accompanied by more modest physiological effects on starch and callose accumulation, even under higher *C*Las-inoculum pressure. It is important to highlight that all citrandarin genotypes tested here are potential citrus rootstocks. However, further studies focusing on the putative benefits of these genotypes used as rootstocks to the citrus canopy, as well as on the effect of *C*Las on their root systems, are necessary to continue to form combative strategies against HLB.

**Supplementary Materials:** The following supporting information can be downloaded at: https://www.mdpi.com/article/10.3390/agronomy12102566/s1. Absolute quantification of CLas dataset (Table S1), representation of experimental setup (Figure S1), standard curves for CLas quantification (Figure S2), and for the starch content (Figure S3) are reported in Supplementary Materials.

**Author Contributions:** Conceptualization: H.D.C.-F., M.C.-Y.; methodology: M.C., T.M.C.; formal analysis: T.M.C. and J.R.; investigation: M.C., T.M.C., and J.R.; original draft preparation: M.C.; T.M.C., H.D.C.-F.; supervision: H.D.C.-F.; funding acquisition: H.D.C.-F. All authors have read and agreed to the published version of the manuscript.

**Funding:** This work was financed by the São Paulo Research Foundation (FAPESP—2020/07045-3). T.M. Cavichioli (FAPESP—2019/06412-5) and M. Curtulo (FAPESP—2021/03989-0) are grateful for their fellowships. H.D.C.F. and M.C.Y. are recipients of research fellowships from CNPq (proc. no. 308164/2021-0 and 313295/2020-4, respectively).

**Institutional Review Board Statement:** Not applicable.

**Informed Consent Statement:** All individuals included in this section have read and agreed to the published version of the manuscript.

**Data Availability Statement:** The data presented in this study are available on request from the corresponding author.

**Acknowledgments:** We thank Luis Fernando Carvalho Silva, Derik Festa, and Franciel Santos Barbosa for their assistance with the plants in the greenhouse.

**Conflicts of Interest:** The authors declare that the research was conducted in the absence of any commercial or financial relationships that could be construed as a potential conflict of interest.

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
