# Peer review of "Temporal Analysis of Candidatus Liberibacter asiaticus in Citrandarin Genotypes Indicates Unstable Infection"

_agronomy, doi:10.3390/agronomy12102566_

Round 1

Reviewer 1 Report

The study is evaluating 14 citrandarin hybrids and their parents for their response to infection with Candidatus Liberibacter asiaticus, the HLB associated pathogen.

The manuscript would have more appropriate as a short communications paper than a full research paper as the experimental design is flawed and the results are week. Instead of using independent replications for each hybrid cultivar, the authors grafted multiple budsticks on one Valencia inoculum source plant; it is unclear what kind of statistical analysis was performed, but it does not seem as if the grafting design was blocked and truly replicated. Moreover, several budsticks died, leaving fewer than 5 replicates in some instances - for testing HLB/CLas responses in citrus a low number of replications does not yield meaningful data. This left little to discuss, rendering the discussion weak. Furthermore, CLas titers alone are not meaningful without any information on plant growth and health. Cultivars can be infected with CLas but still be tolerant and not develop any disease symptoms. I therefore suggest shortening the manuscript to resubmit as a short communication or conduct an additional, well-designed confirmatory study.

Other comments:

General

       Please consult with a native speaker; although the language is ok overall, there are some odd phrases (e.g., line 57-60).

Introduction:

-        The objectives are not stated.

Materials and methods

       Line 84: point iii is unclear

       Spell out “Pt cv Rub” the first time

       The term “source” plant is confusing/misleading; obviously, the infected Valencia plants are a CLas source, but they are the recipients of the budsticks; maybe “recipient plants” or “recipient (source) plants” would be more appropriate

       Line 94: add “budsticks” after “received”

       Specify the composition and vendor of the PlantMax medium

       Were plants kept under natural lighting?

       As mentioned above, the experimental design is flawed, and the statistical analysis is unclear

       How large were the grafted hybrids by the end of the study?

       Why were dark green colored leaves collected (line 108)? Were there no HLB symptoms?

       Why is no information on HLB symptoms and plant growth provided? This is more important than the Ct-values

       Line 120: Are you sure you had 100ng/ul of DNA? This seems an awful lot; maybe you mean per reaction?

       Starch quantification: Some brief details on the method would be helpful, especially since the method article cited is not in English.

Results

-        The way the results are presented reflects the poor experimental design. Table 1 shows fluctuations in the source/recipient plants and as the experimental hybrids were not grafted as independent replicates AND with only 3-5 pseudo replications, the results are not meaningful.

-        Table 1 and the calculated percentages based on a maximum of 5 budsticks are not meaningful.

-        Starch and callose accumulation should have been correlated with CLas titers. Also, there could be varietal differences, regardless of infection.

-        Figure 1 A is not helpful. It would be better to present varietal differences and correlate starch with Ct-values

Discussion

-        Line 277: use “HLB”, not “HLB disease”

-        Line 282: there are no “biostimulators” (it should be “biostimulants”) and the reference is not about biostimulants anyway

-        Line 296-297: calculating percentages based on 3-5 plants inflates the results

-        Line 306-312: the authors describe many potential pitfalls associated with conducting CLas graft transmission experiments – why did they conduct such a poorly designed experiments if they are aware of the pitfalls?

-        Line 319: what is “a certain degree of differences”?

-        Line334-355 are not relevant to the study (one sentence would have been sufficient)

Author Response

Dear reviewer, 

the authors thank you for your valuable contribution. We try to clarify all the doubts and answer all the questions. See in attached file, please.   

Reviewer 2 Report

Overall, this is a decent study. This reviewer considers the title slightly inappropriate since transiency is not discussed. Overall, the results are interesting and reasonable but there are a few points where the discussion could be improved. Also, the anomalous H106 could be discussed a little more, although this reviewer is not sure exactly how. Its behavior just seems weird.

66-78: Is this actually the conclusion stated in the introduction?

88-89: Were the valencia orange plants seedlngs?

189-196: Table 3 apparently based upon the criterion stated in 160-164 (ie, Ct < 32). Why are valus not reported? How does variation in Ct in the citrandarin scions compare to the valencia rootstock?

218: H109 in text should be H106 (very important point).

244-248: Table 3 not aligned correctly with text. Are these values averages? Not clear how this associates with Table 2. 

250-275: Figure 1A: Not clear what the 2 bars for each selection represent: Low and High values?

Author Response

(The authors gave the same response as above.)

Round 2

Reviewer 1 Report

The revised manuscript is improved, but there are still a few things that need to be addressed:

 Line 33: Change fruit “drops” to fruit “drop”.

Line 67-68: I is unclear what is meant by “contrasting for HLB resistance”. Better to change it to something like “This work aimed to investigate the colonization of CLas in 14 different citrandarin hybrids regarding their resistance to HLB”.

Line 90: Change “pinus” to “pine”

Line 91: provide details for Terra do Paraiso (city, state, country)

Line 92: Be consistent in the use of “14” and “fourteen”.

Line 92-93: Sentence is confusing (“Branches … (qPCR).” Change to “Branches from the HLB-affected Valencia plants …”

Line 94: change “All the” to “All”

Line 96 and everywhere else: change “HLB-diseased” to “HLB-affected recipient plants” or “CLas-infected budstick recipient plants”, or “CLas-infected recipient plants”, or something more appropriate.

Line 143: Change “Glucose” to “glucose”

Line 144-145: Change “commercially available kit glucose PAP Liquiform” to “Glucose PAP Liquiform kit”

Table 1: What does “SP” stand for?

Line 186: Wouldn’t it be “side-grafting” not “top-grafting”?

Line 196: Now the term “Valencia CLas source plants” is used, this is confusing – be consistent.

Line 329: change sentence to “…infected or not infected plants accumulated…”

Line 330: change “citradarin” to “citrandarin”

Lines 350-361: These details on the putative transcriptional mechanisms associated with HLB resistance are irrelevant to the study - this paragraph needs to be shortened.

Line 371: The term “differential tolerance” does not make any sense. Please clarify/revise.

Other: Add information on how the grafted hybrids looked (did they display HLB symptoms?) and how large they were at the end of the study/sampling time in the results and discuss. If you did not see any foliar symptoms in the grafted hybrids, that is important and needs to be documented.

For future studies, it is strongly recommended to improve the experimental design and include more biological replications.

Author Response

See the attached file, please. All the questions were answered one by one. 
